# Prediction of Diabetic Macular Edema Using Knowledge Graph

**DOI:** 10.3390/diagnostics13111858

**Published:** 2023-05-26

**Authors:** Zhi-Qing Li, Zi-Xuan Fu, Wen-Jun Li, Hao Fan, Shu-Nan Li, Xi-Mo Wang, Peng Zhou

**Affiliations:** 1Academy of Medical Engineering and Translational Medicine, Tianjin University, Tianjin 300072, China; 2Tianjin Medical University Eye Hospital, Tianjin 300392, China; 3School of Optometry & Eye Institute, Tianjin Medical University, Tianjin 300392, China; 4Tianjin Branch of National Clinical Medical Research Center for Eye, Ear, Nose and Throat Diseases, Tianjin 301999, China; 5Tianjin Key Laboratory of Retinal Function and Diseases, Tianjin 300383, China; 6Tianjin Key Laboratory of Intelligent Traditional Chinese Medicine Diagnosis and Treatment Technology and Equipment, Tianjin 300072, China; 7Tianjin Medical University Zhu Xian Yi Memorial Hospital, Tianjin 300070, China; 8Department of Biomedical Engineering, Washington University in St. Louis, St. Louis, MO 63130, USA; 9Tianjin Hospital of Integrated Traditional Chinese and Western Medicine, Tianjin 300102, China; 10School of Precision Instrument and Opto-Electronics Engineering, Tianjin University, Tianjin 300072, China

**Keywords:** diabetic macular edema, disease prediction, knowledge graph, neo4j, personalized prediction, clinical decision support system

## Abstract

Diabetic macular edema (DME) is a significant complication of diabetes that impacts the eye and is a primary contributor to vision loss in individuals with diabetes. Early control of the related risk factors is crucial to reduce the incidence of DME. Artificial intelligence (AI) clinical decision-making tools can construct disease prediction models to aid in the clinical screening of the high-risk population for early disease intervention. However, conventional machine learning and data mining techniques have limitations in predicting diseases when dealing with missing feature values. To solve this problem, a knowledge graph displays the connection relationships of multi-source and multi-domain data in the form of a semantic network to enable cross-domain modeling and queries. This approach can facilitate the personalized prediction of diseases using any number of known feature data. In this study, we proposed an improved correlation enhancement algorithm based on knowledge graph reasoning to comprehensively evaluate the factors that influence DME to achieve disease prediction. We constructed a knowledge graph based on Neo4j by preprocessing the collected clinical data and analyzing the statistical rules. Based on reasoning using the statistical rules of the knowledge graph, we used the correlation enhancement coefficient and generalized closeness degree method to enhance the model. Meanwhile, we analyzed and verified these models’ results using link prediction evaluation indicators. The disease prediction model proposed in this study achieved a precision rate of 86.21%, which is more accurate and efficient in predicting DME. Furthermore, the clinical decision support system developed using this model can facilitate personalized disease risk prediction, making it convenient for the clinical screening of a high-risk population and early disease intervention.

## 1. Introduction

According to statistics from the World Health Organization (WHO), there are approximately 285 million people worldwide who suffer from visual impairment, and 4.8% of these cases are caused by diabetic retinopathy (DR) [1,2]. Diabetic macular edema (DME) is a serious complication of diabetes that affects the eye and is an advanced symptom of DR. It is one of the main causes of irreversible vision loss in diabetic patients [3], affecting approximately 1/15 diabetic patients globally [4]. As the number of diabetic patients continues to increase annually [5], DME imposes a significant economic burden on both society and families. Since DME can occur at any stage of DR [6], early control of the associated risk factors is a crucial strategy to reduce its incidence. Therefore, it is critical to screen a high-risk population for DME and provide early clinical intervention.

The disease prediction model can comprehensively consider the multiple factors that influence the occurrence of DME [7]. By predicting the probability of DME, the model can assist in the clinical screening for a high-risk population and aid in the early intervention for and treatment of patients to prevent further disease progression.

Over the past few years, researchers in both domestic and international contexts have conducted extensive studies on the clinical treatment of DME [8]. However, the absence of comprehensive risk factor prediction and evaluation has hindered the application of disease prediction models, to a certain extent, in the prediction and analysis of DME, thereby delaying early treatment for affected patients.

At present, disease prediction models are primarily based on data mining and machine learning techniques. However, these models may encounter challenges in meeting the requirements of personalized disease prediction and predicting diseases when feature values are missing. To address these issues, a knowledge graph has emerged as a promising solution, as it can efficiently process multi-source heterogeneous data, conduct correlation analyses, and accurately predict disease outcomes. Therefore, a knowledge graph has become increasingly popular in the medical and healthcare fields.

A knowledge graph is a repository of various types of entities, concepts, and their semantic relationships. It consists of a collection of knowledge, with each piece represented as a triple (entity, relation, and entity). The significance of this research lies in the rich background knowledge it provides for semantic matching and machine learning. Currently, in addition to Google’s Knowledge Graph, there are several other high-quality and widely used open knowledge graphs in the world. These include DBpedia, Wikidata, ConceptNet, and Microsoft Concept Graph, which cover multiple languages and a diverse range of fields. Additionally, there is OpenKG, a Chinese open knowledge graph platform.

A knowledge graph [9] is a type of knowledge base that utilizes semantic retrieval techniques to collect information from various sources and improve retrieval quality. It has been widely applied in clinical auxiliary decision support systems, such as the Nanjing Dajing traditional Chinese Medicine (TCM) Clinical Intelligent Auxiliary Decision Support System, the Ancient and Modern Medical Record Cloud Platform, the Daosheng Medicine-TCM Data Intelligent Service System, the TCM Artificial Intelligence Auxiliary Diagnosis and Treatment Software, and the Qi-Huang Data AI Workstation. By leveraging artificial intelligence technologies and clinical datasets from Hospital Information Systems (HIS), these systems can obtain high-quality structured data through online integration and processing and subsequently train and apply intelligent models using data mining and analysis techniques.

The life cycle of a knowledge graph is shown in Figure 1, which can be divided into an ontology layer and an entity layer. The ontology layer represents the core of a knowledge graph, and its main content is the data model of the knowledge classes. This model is presented as concepts and relationships, including the hierarchical structure and relationship definitions of the knowledge classes, such as entities, relationships, and attributes. The entity layer is responsible for storing specific data information, which is represented and stored as a directed graph in the form of triples (entity, relationship, and entity). Entities typically represent specified objects or things, relations represent the connection relationships between entities, and attributes and their values represent the parameters of an entity or relationship.

**Multi-source heterogeneous data:** The data sources for a knowledge graph can include structured data (such as databases or canonical form data), semi-structured data (such as tables), and unstructured data (such as text and words).

**Knowledge extraction:** This process involves extracting entities, relationships, and attributes from multi-source heterogeneous data. Traditional rules, machine learning, and deep learning methods are often used to extract the required knowledge. Currently, the most popular method is CNN [10], which extracts lexical and sentence features for knowledge extraction. Socher et al. [11] used RNN to obtain vector features of sentences, which improved the performance of relation extraction. Zhang et al. [12] proposed a Bi-LSTM method to obtain information between words for relation classification.

**Knowledge fusion:** This process involves disambiguating, processing, and integrating heterogeneous and diverse knowledge from different data sources in the same framework, thus achieving the fusion of data and information from multiple perspectives. Currently, the technology for knowledge fusion is mainly divided into ontology fusion and data fusion. Ontology fusion involves integrating multiple heterogeneous ontologies from data sources into a unified ontology and establishing mapping rules between multiple ontologies, so information can be transferred between different ontologies. Data fusion involves using methods such as entity merging, entity alignment, and entity attribute fusion to achieve data unification and standardization.

**Knowledge representation:** This is a way of describing knowledge that involves transforming massive real-world information into structured data using information technology. Currently, the common methods of knowledge representation include XML [13], RDF [14], RDFS [15], and OWL [16], among others. XML represents knowledge in the form of documents and allows users to mark data and define data types. RDF uses a unified standard “subject-predicate-object” triple to describe entities and relationships, which can also be expressed as a directed graph structure. RDFS can be used as an extension of RDF with schema definitions and simple constraint rules for RDF entities, attributes, and relationships. OWL is based on RDFS, adding a predefined vocabulary to describe the characteristics of resources.

**Knowledge storage:** This component is used to manage and store knowledge. The main methods for storing knowledge include an RDF database, a traditional relational database, and a graph database. Among them, a graph database has become the mainstream method for knowledge storage, as it represents data as nodes and edges and clearly shows the dependencies between data nodes. Graph query languages support various graph mining algorithms, and popular graph databases include Neo4j [17], JanusGraph [18], HugeGraph [19], etc.

**Knowledge reasoning:** This process infers unknown relationship information based on existing entity relationship information, which further enhances the completeness and usefulness of a knowledge graph. There are several methods of knowledge reasoning, including reasoning based on logical rules, reasoning based on distributed feature representation, and reasoning based on a neural network.

**Knowledge application:** A knowledge graph is widely applied in semantic retrieval, knowledge question answering, recommendation systems, and decision making. A semantic search is used to understand user retrieval needs at the semantic level and search for matching resources. Answering a knowledge question involves converting user input questions into objective entity concepts within the knowledge base, which is achieved through structured query mapping via natural language methods in order to obtain the answer. Recommendation systems and decision making involve providing multi-level decision support and knowledge services for decision-making systems by enhancing the mining ability of recommendation algorithms in the knowledge graph, taking into consideration the personalized preferences of users.

At present, there is a lack of research on disease prediction models in the field of DME, and the development of a DME disease prediction model is of great significance for the early screening of a high-risk population. In this context, this study used knowledge graph technology to implement clinical auxiliary disease prediction functions to meet clinical needs. The clinical data collected by the Tianjin Medical University Eye Hospital were used as the research object, and a DME disease prediction model based on the knowledge graph [20] was proposed. By comprehensively analyzing the different influencing factors, the model was able to obtain the incidence probability of a DME disease.

In addition, a clinical auxiliary decision support system was developed to enable personalized disease prediction analysis, which can effectively assist in the clinical screening of a high-risk population for DME, provide technical support for disease prediction, and serve as a reference for innovative applications of disease prediction models. The innovations are as follows.

A statistical analysis was conducted on the clinical data to identify the high-risk factors of DME for clinical reference and early intervention.Personalized disease prediction model was developed, which comprehensively considered the disease influencing factors and analyzed the disease probability of the target population.Knowledge graph technology was utilized to construct the disease prediction model, facilitating data updating, maintenance, and iterative improvements.A clinical decision support system was developed to promote the implementation and usage of this model in clinical practice, providing a technical foundation for an innovative application of the knowledge graph.

## 2. Literature Review

Studies showed that the risk factors for DME [21] include hyperglycemia, duration of diabetes, hypertension, hyperlipidemia, renal dysfunction, pregnancy, the level of cytokine VEGF, heredity, etc. Additionally, other factors such as a history of cataract surgery, insulin use, sleep apnea syndrome, anemia, and ocular inflammation were also identified.

In addition, studies showed that the level of β-collagen degradation products (β-CTx) is significantly correlated with DME in female patients with type 2 diabetes, and high serum calcium can also aggravate macular edema in patients with type 2 diabetes. Moreover, it is widely accepted that DME is caused by oxidative stress induced by continuous hyperglycemia and the accumulation of inflammatory cytokines and vascular endothelial growth factor (VEGF), leading to the disruption of the blood–retinal barrier (BRB) [22,23]. Due to the complex pathogenesis of DME, controlling the systemic factors that contribute to DME progression is key to reducing its morbidity.

At present, disease prediction models in the medical big data environment mainly use data mining and machine learning methods to predict unknown data. In medical diagnosis analysis, the characteristic values of disease types are extracted from case examination items, and the probability of different disease types with different characteristic values is analyzed to construct disease prediction models.

Puchao, H. et al. [24] used SVM, decision tree C5.0, and ANN methods in a data mining model to construct a lung cancer risk prediction model.

Xing, W. et al. [25] used the decision tree classification method model to test and analyze historical cases and predict the prevalence probability of diseases such as cold and cough.

Alourani, A. et al. [26] used a model based on a deep neural network to predict the mortality of patients in health cloud data.

Raju, M. et al. [27] used machine learning methods to capture the clinical symptoms that may have an EHR history in potential pre-glaucoma patients, to achieve early intervention and preventive treatment of the disease.

However, these methods require sufficient feature data selection to complete disease prediction, which is challenging when faced with missing feature values. To address this issue, knowledge graph reasoning can infer unknown relationships or facts from the existing relationships or facts using any number of entity feature values and apply simple rules or statistical features for reasoning. By combining manually defined logical rules with various probabilistic graphical models, knowledge reasoning can be performed on the constructed logic network, which plays a crucial role in the fields of medical diagnosis, disease prediction, medical treatment, and medical standardization. A knowledge graph has a wide range of applications in TCM clinical, TCM basic, TCM health care, and other fields, including semantic retrieval, intelligent question and answer, decision support, etc. It lays a solid foundation for the intelligent application of TCM and can also be extended to other disease prediction fields.

Tong, Y. et al. [28] constructed a knowledge graph for the field of TCM to achieve an effective integration of TCM knowledge resources.

Li, Y. [29] took ILP as the basic algorithm framework, combined it with the characteristics of TCM diagnosis and treatment to form a knowledge base of TCM clinical syndrome diagnosis rules, and used the relevant weight training algorithm to learn the weights of the learned rules in MLN.

Yingying, Z. [30] constructed a TCM knowledge graph, applied it to the field of TCM diagnosis and treatment, and designed a tongue image diagnosis and treatment system based on the knowledge graph. According to the TCM diagnosis process, the final syndrome diagnosis results and treatment plans were given through the symptoms and tongue body photos entered by users.

Ziqiang, Z. [31] constructed a knowledge graph of the TCM medical cases of chronic kidney disease to perform representation learning of the CKD TCM medical case knowledge graph and reasoning of the CKD TCM medical case knowledge graph, realizing the learning and reasoning of the knowledge graph.

Dan, Y. et al. [32] used the construction method of a knowledge graph and graph search pattern to construct the pattern layer and data layer design of the knowledge graph with TCM classics prescriptions as the research object and designed a knowledge retrieval framework of classics’ prescriptions by using Cypher language in Neo4j.

Kai, Z. et al. [33] used the Neo4j graph database to build a small knowledge graph based on the Guizhu decoction prescriptions in “Shang Han Lun” and realized the visual analysis and retrieval functions of the syndromes, prescriptions, and drugs of the Guizhu decoction prescriptions.

Wenlong, G. [34] realized the construction (knowledge fusion, knowledge acquisition, knowledge storage, and knowledge reasoning) and visualization framework of a TCM prescription knowledge graph.

Fan, L. [35] proposed a standardized system process for the construction of a knowledge graph in the field of experience inheritance of famous and old Chinese medicine.

Dan, Z. [36] constructed a knowledge graph of the rules of fatty liver disease syndrome treatment using famous old Chinese medicine and realized a summary of the rules of syndrome treatment by medical case collection, data aggregation, data preprocessing, and data analysis and mining methods (online analysis technology, the complex network method, and the FP-Growth algorithm).

By presenting the interconnection between multi-source and multi-domain data through a semantic network, a knowledge graph enables cross-domain modeling and querying [37] and allows for the expansion of data on demand without disrupting the existing data structure. It has found extensive applications in semantic retrieval [38], intelligent question answering [39], decision support [40], and other fields.

Knowledge reasoning refers to the process of using existing knowledge, rules, and constraints to acquire new knowledge. This process involves reasoning about unknown information based on known information [41]. For example, tasks such as recommendation computing [42], causal analysis, and query question answering can be reduced to the problem of reasoning about the relationships between entities. Knowledge graph reasoning is a technique that can be used to perform tasks such as attribute completion, relation prediction, error checking, question expansion, and semantic understanding [43]. There are several methods for performing knowledge reasoning, including rule-based reasoning, distributed reasoning, and neural-network-based reasoning [44]. Rule-based knowledge reasoning is based on the connection rules or statistical features of entities. The rules are obtained by the statistical learning of the whole knowledge graph, and the rules are improved and optimized on this basis, which can be applied in the field of DME disease prediction.

## 3. Materials and Methods

Initially, the dataset was split into a training set and a test set with an 8:2 ratio. Subsequently, the entities and relationships of knowledge graph were constructed through statistical analysis and data preprocessing.

On this basis, the correlation enhancement algorithm was applied to refine the weight attributes of the connecting edges in the knowledge graph. In addition, the generalized closeness method was utilized to enhance the disease prediction model, enabling the classification of influencing factors among different target populations. Ultimately, expert suggestions informed by clinical practice were provided based on the obtained results.

As a result, 3 DME disease prediction models were analyzed and verified based on link prediction evaluation indicators, including AUC, accuracy, and ranking score, to select a more accurate and efficient disease prediction model.

The disease prediction model proposed in this study is shown in Figure 2, its activity sequence is shown in Figure 3, and the disease prediction process is as follows.

**Data preprocessing:** The original clinical data were collected, preprocessed, and subjected to statistical analysis. Initially, clinical diagnostic data and basic information of cases were classified based on clinical standards, and then data related to biochemical and non-biochemical indicators were processed separately to identify a limited number of influencing factor entities. The preprocessed data were subjected to statistical analysis, which revealed the high-risk factors associated with DME in the clinical data.

**Model training:** After the preprocessed dataset was split into training set and test set, the model was trained on the data of the training set, and the knowledge extraction task was completed by means of entity extraction, attribute extraction, and relationship extraction. A medical knowledge graph was constructed, the disease prediction model was constructed by using the correlation enhancement algorithm and the generalized paste progress method, and the model training was completed.

**Model testing:** The knowledge reasoning algorithm based on the knowledge graph was used to predict the disease on the test set and the ontology rules constructed by the knowledge graph. It includes disease prediction models based on statistical rule reasoning, based on correlation enhancement, and based on improved correlation enhancement.

**Analysis and evaluation:** Three models were verified and analyzed according to the parameters of the link prediction evaluation indicators of the knowledge graph, including AUC, precision, and ranking score.

### 3.1. Dataset

This study utilized clinical data from 507 cases, which were collected by the hospital information system of Tianjin Medical University Eye Hospital. Specifically, the data comprised 261 cases of DME and 246 cases of non-DME.

### 3.2. Statistical Analysis and Preprocessing of Data

The influencing factors of the disease were based on 39 categories, as shown in Table 1. These factors were preprocessed, classified, and subjected to statistical analysis, and the quantitative numerical data were transformed into qualitative data that could be interpreted.

The processing of biochemical data was divided according to the normal value range or interval stage, such as fasting plasma glucose, as shown in Table 2. The processing of non-biochemical data was divided according to 0/1, such as family history of diabetes, as shown in Table 3. The statistical classification results of the data are shown in Table 4.

According to the results of data analysis after data processing, 39 categories of influencing factors were divided into 116 qualitative interpretable value ranges. Using the statistical rule reasoning method based on knowledge graph, the classification statistical results of influencing factors were used as the main basis for the weight setting of the connection edges in the process of constructing DME knowledge graph, which provides data support for the comprehensive evaluation of influencing factors in the process of disease prediction.

### 3.3. Construction of Knowledge Graph

After statistical classification and preprocessing of data, a medical knowledge graph was constructed, consisting of 2 types of nodes: “whether or not the disease” and “116 DME disease influencing factors”, connected by 232 relationships. The weight attributes of the relationships are added later by computing the weight calculation results, which completes the task of knowledge completion [45].

Figure 4 shows the schematic diagram of DME disease prediction model based on improved correlation enhancement. Through data preprocessing and statistical analysis of the basic information of the case, biochemical indicators, and influence factors of non-biochemical indicators, the DME data in the training set were input into the knowledge graph as historical case data. In the process of model validation, the similarity between the input personalized diagnosis data results and the historical cases was compared. The weight was used to represent the impact of each influencing factor on the disease, and the correlation enhancement and generalized progress method were used to further obtain the calculation results of DME disease probability after feature fusion.

### 3.4. Disease Prediction Models

To achieve a comprehensive assessment of the impact of multiple influencing factors on disease risk, the DME knowledge graph was constructed using weight as the connection edge attribute between the disease node and the influencing factor node. The statistical analysis results of the training set data were utilized as the standard for weight setting during the construction of the knowledge graph [46]. The clinical dataset was randomly divided into training set and test set at a ratio of 8:2, including 406 cases of training set (203 cases of DME and 203 cases of non-DME), and 101 cases of test set data contained 58 cases of DME clinical data. The statistical characteristics of the training set data were used as attributes for the knowledge graph connection edge. Three weight setting methods are proposed as follows.

Weight 1 is the frequency of cases in which the *i* disease influencing factor is connected to the *j* disease node, which is the frequency in the sum of the case frequencies of all node pair connection relationships, where *i* = 1… 39 corresponding to 39 disease influencing factors, *j* = 0… 1, 0 denotes non-DME, 1 denotes DME, and $n_{ij}$ denotes frequency under corresponding conditions.
(1)$${weight_{1}}_{ij}=\frac{n_{ij}}{\sum_{i=1}\sum_{j=0}n_{ij}}$$

Weight 2 is the case frequency connected by the *i* disease influencing factor and the *j* disease node, the frequency ×100 in the sum of case frequencies connected by the node with or without disease, and the influencing factor under this influencing factor, where *i* = 1… 39 corresponding to 39 disease influencing factors, *j* = 0/1. 0 represents non-DME, 1 represents DME, and $n_{ij}$ represents the frequency in the corresponding condition.
(2)$${weight_{2}}_{ij}=\frac{n_{ij}}{\sum_{j=0}n_{ij}}\times 100$$

Weight 3 is combined with the correlation enhancement algorithm based on the results of weight 2 setting, that is, weight 2× correlation enhancement coefficient, where *i* = 1… 39 corresponding to 39 disease influencing factors, *j* = 0/1. 0 represents non-DME, 1 represents DME, $n_{ij}$ represents the frequency under the corresponding condition, *z* = 1… 5, and $\alpha_{z}$ denotes the correlation enhancement coefficient (Table 5).
(3)$${weight_{3}}_{ij}=\frac{n_{ij}}{\sum_{j=0}n_{ij}}\times 100\times \alpha_{z}$$

The correlation enhancement coefficient indicates that for weight 2 calculation results, if the frequency of an influencing factor node in DME and non-DME datasets is not significantly different (i.e., the frequency range is closer to 50%), it implies that the factor has less effect on DME disease, resulting in a smaller correlation enhancement coefficient. Conversely, a larger correlation enhancement coefficient indicates a stronger influence of the factor. These coefficients are ranked based on the frequency range priority, with 1 being the highest and 5 being the lowest.

The statistical analysis features of the training set were applied to the knowledge graph to complete the disease prediction task, and the calculation result of weight 1 was used as the attribute value of the knowledge graph connection edge to construct a DME disease prediction model based on statistical rules. On the basis of setting the attribute values of the connected edges of the knowledge graph with weight 2, the correlation enhancement algorithm was used to set weight 3 to further improve the correlation between the influencing factors and the disease, and a DME disease prediction model based on correlation enhancement was constructed. In addition, the method of generalized closeness [47] was used to divide the disease prediction results into interpretable intervals, the DME disease prediction formula was obtained, as shown in Equation 4, and the DME disease prediction model based on improved correlation enhancement was constructed.
(4)$$T\left(M^{+i},M^{0}\right)=1-\frac{1}{W_{max}}\cdot \frac{1}{n}\sum_{i=1}^{n}\left|W_{max}-W_{i}\right|\cdot \left|\mu_{s}\left(\mu_{i}^{+}\right)-\mu_{s}\left(\mu_{i}^{0}\right)\right|$$

$T\left(M^{+i},M^{0}\right)$ values are used for similarity discrimination to calculate DME disease prediction results, where $M^{+i}$ is the dataset of the *i*th disease influencing factors in the knowledge graph (*i* = 1, 2…). $M^{0}$ is the current input dataset. μ values represents the data connection relationship in the knowledge graph. If there are data connected with influencing factors at the same time, that is, $\mu_{s}\left(\mu_{i}^{+}\right)=\mu_{s}\left(\mu_{i}^{0}\right)$, the probability of DME disease in the target population is 1. The weight $W_{i}$ of the edge is the result of setting weight 3, and the maximum value of the weight $W_{max}$ = 500. If the edge weights $W_{i}$ = $W_{max}$, the two nodes are fully weighted connected, so $\left|\mu_{s}\left(\mu_{i}^{+}\right)-\mu_{s}\left(\mu_{i}^{0}\right)\right|$ = 0; otherwise, $\left|\mu_{s}\left(\mu_{i}^{+}\right)-\mu_{s}\left(\mu_{i}^{0}\right)\right|$= 1.

Take the predicted DME probability $P\in [{Predict}_{min},{Predict}_{max}]$ calculated by Equation (4), and scale the range $\left[{Predict}_{min},{Predict}_{max}\right]$ to the clinically interpretable range ${[P}_{min},P_{max}]$ = [0, 100]. Thus, the probability of DME predicted by the proposed disease prediction model for the target population was obtained.
(5)$$P_{DME}=P_{min}+\frac{P_{max}-P_{min}}{{Predict}_{max}-{Predict}_{min}}\times \left(P-{Predict}_{min}\right)$$

The range of ${[P}_{min},P_{max}]$ was used as the range of DME disease prediction results. After discussion with clinical experts, the probability of DME was divided into four categories, and relevant suggestions were given, as shown in Table 6.

## 4. Results

The partial results of the weight setting based on correlation enhancement are used as attributes between the disease nodes and influencing factor nodes in the knowledge graph (Table A1), and the results are filled into the knowledge graph.

After data processing and data analysis, the top six influencing factors with the highest weights in the DME data were Scr, DR, urine protein, 24 h PRO, 24 h MALB, and FIB. The statistical results are shown in Figure 5.

A DME disease prediction analysis was performed on the test set of the DME dataset using the DME disease prediction model based on improved correlation enhancement, and the results are shown in Table 7.

According to the validation results of the training set, taking a disease probability of 70% as the standard for predicting DME disease, the DME case data in the test set were analyzed and verified, and the DME disease prediction results shown in Table 8 were obtained. According to the DME disease prediction model based on improved correlation enhancement, the disease probability of the 58 cases of DME data in the test set was predicted. The results showed that the disease probability was mostly 70–85%, which was consistent with the expected results.

Table 9 presents the evaluation results of the three disease prediction models, namely, AUC [48], precision [49], and ranking score [50], based on the link prediction evaluation parameters of the knowledge graph reasoning. These results were obtained through analysis and verification.

The AUC measures the accuracy of an algorithm as a whole. Given a prediction algorithm, for each unknown node will be given a value of the likelihood of existence. After training, the algorithm obtains the similarity value of each pair of nodes in the knowledge graph network, and a degree greater than 0.5 indicates that the degree of the algorithm is better than the random selection algorithm.
(6)$$AUC=\frac{n_{1}+0.5n_{2}}{n}$$

The precision is the proportion of accurate predictions among the top L predicted nodes. If the probability value of the connection is arranged from large to small, and m nodes in the top L are in the test set, then the accuracy is defined as Equation (7).
(7)$$\begin{array}{c} Precision=\frac{m}{L} \end{array}$$

The ranking score considers the position of the nodes in the test set in the final ranking, and the smaller the ranking score is, the better the prediction effect of the algorithm is. *H* is the collections of unknown nodes, $r_{e}\in E^{p}$ denotes test node e in sorting ranking, and the ranking score of the test nodes is ${RS}_{e}=\frac{r_{e}}{\left|H\right|}$. By traversing all the nodes in the test set, the system ranking score is obtained.
(8)$$RS=\frac{1}{\left|E^{p}\right|}\sum_{e\in E^{p}}\frac{r_{e}}{\left|H\right|}$$

The link prediction evaluation index parameters of the knowledge graph reasoning were used to analyze and verify the three disease prediction models. Based on the evaluation results of the three models, the algorithm using correlation enhancement demonstrated superior performance than the statistical rule reasoning, with a 38.65% higher AUC and a 38.15% higher precision. The algorithm based on improved correlation enhancement showed even better performance, with a 27.95% higher AUC and a 28.45% higher precision compared to the previous method. The proposed DME disease prediction model based on improved correlation enhancement can more accurately and effectively predict the disease by comprehensively evaluating the diagnostic data of the cases. The model can facilitate the clinical screening of the high prevalence of a DME disease, provide references for early disease intervention, and achieve the expected effect.

A clinical decision support system was developed using the improved DME disease prediction model, which enables personalized disease risk prediction by allowing users to input any number of diagnostic data results. Figure 6 illustrates the flowchart of this system.

For example: The influencing factors were “female, 60–69 years old, BMI is 24–28 $kg/m^{2}$, Waist-to-hip ratio of central obesity, duration of diabetes is 10–19 years, family history of diabetes, hypertension, hyperlipidemia, diabetic nephropathy, anemia, no coronary heart disease, no cerebral infarction, diabetic peripheral neuropathy, no pre-cervical dark spots, diabetes. The stage of retinopathy is PDR, no history of cataract surgery, no history of retinal laser photocoagulation, smoking history, no history of alcohol consumption, use of insulin, glycosylated hemoglobin > 9%, mean platelet volume 7–11 fL, erythrocyte sedimentation rate >20 mm/h, triglyceride 0.45–1.7 mmol/L, total cholesterol 2.9–5.18 mmol/L, high density lipoprotein cholesterol 1.04–1.55 mmol/L, low density lipoprotein cholesterol 3.37–4.12 mmol/L, D-dimer 0–0.5 mg/L, fibrinogen 2–4 g/L, urea 2.5–7.5 mmol/L, creatinine <88.4 μmol/L, uric acid 89–357μmol/L, estimated glomerular filtration rate 80–120 mL/min, total protein < 60 The probability of DME in the users with g/L, albumin < 40 g/L, urinary protein (+), 24-h urinary protein < 0.15 g, and 24-h urinary microalbumin < 15 mg”. For these people, the probability of DME is 86.65%, and a special examination is recommended.

## 5. Discussion

A knowledge graph allows for disease prediction even in the case of missing clinical data values. This method is more universally applicable and significant compared to traditional data mining algorithms, as it can complete disease prediction tasks with small amounts of data using simple statistical rule reasoning. The DME disease prediction model based on improved correlation enhancement proposed in this study defines a simple and effective rule-based reasoning algorithm, which realizes a more accurate and effective reasoning application. It not only provides a reference for the clinical screening of a DME high-risk population but also provides research ideas and methods for the early prediction and intervention of other clinical major special diseases. This model can comprehensively evaluate disease influencing factors using any number of objective diagnostic data and tolerate incomplete data. Compared to traditional data mining classification algorithms, it facilitates model optimization, update and iteration, and can assist in clinical screening for high-risk populations for diabetic macular edema, providing expert advice based on different disease probabilities and, thus, contributing to clinical disease prediction and early intervention.

The proposed DME disease prediction model uses a knowledge graph to statistically analyze clinical data, classify the 39 types of DME disease influencing factors, and construct a medical knowledge graph using three weight-setting methods for the knowledge graph connection edge. Based on this, the disease prediction formula was derived using the generalized closeness degree method and improved using correlation enhancement. Three DME disease prediction models were tested, and the precision of the disease prediction model based on improved correlation enhancement was increased by 28.45% compared to the algorithm before improvement, indicating the model’s better performance in completing the disease prediction task and providing technical methods for the intelligent prediction of clinical diseases.

In this study, the proposed DME disease prediction model comprehensively evaluates disease probability based on basic clinical data, disease history, medical test results, and other factors, resulting in a clinical decision support system for disease prediction visualization [51]. This system is useful in promoting and applying the disease prediction model in clinical practice. The study plans to collect more clinical objective data in the future to update the model and further improve its accuracy of disease prediction. Additionally, this study’s findings can serve as a reference for the prediction of other major clinical diseases such as hypertension, diabetes, coronary heart disease, and others, providing support for early disease screening and intervention. Furthermore, the model can offer decision-making technical support for TCM syndrome differentiation, disease diagnosis, disease treatment, and physical identification.

## Figures and Tables

**Figure 1 diagnostics-13-01858-f001:**
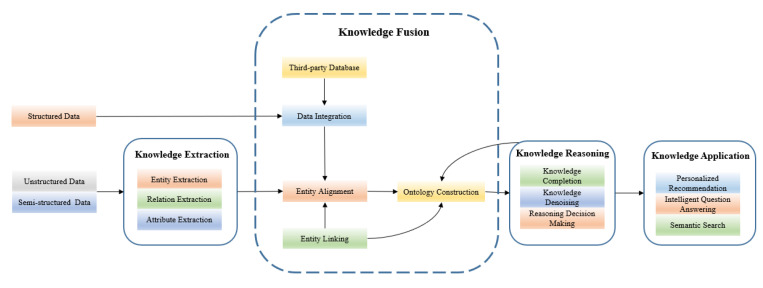
The life cycle of knowledge graph.

**Figure 2 diagnostics-13-01858-f002:**
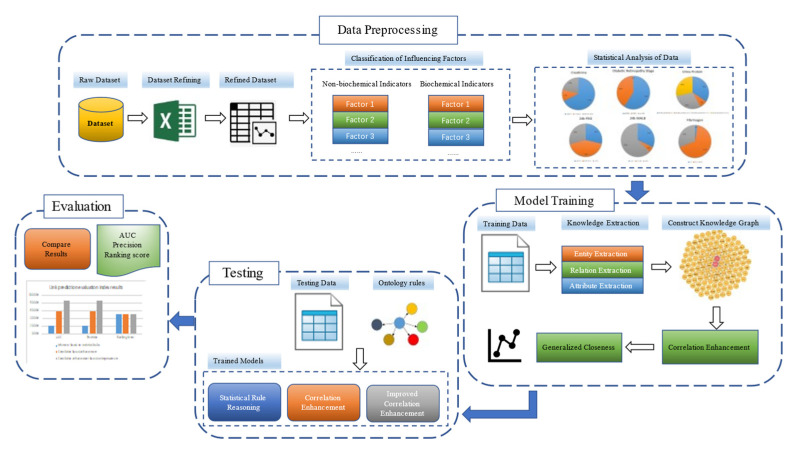
DME disease prediction model.

**Figure 3 diagnostics-13-01858-f003:**
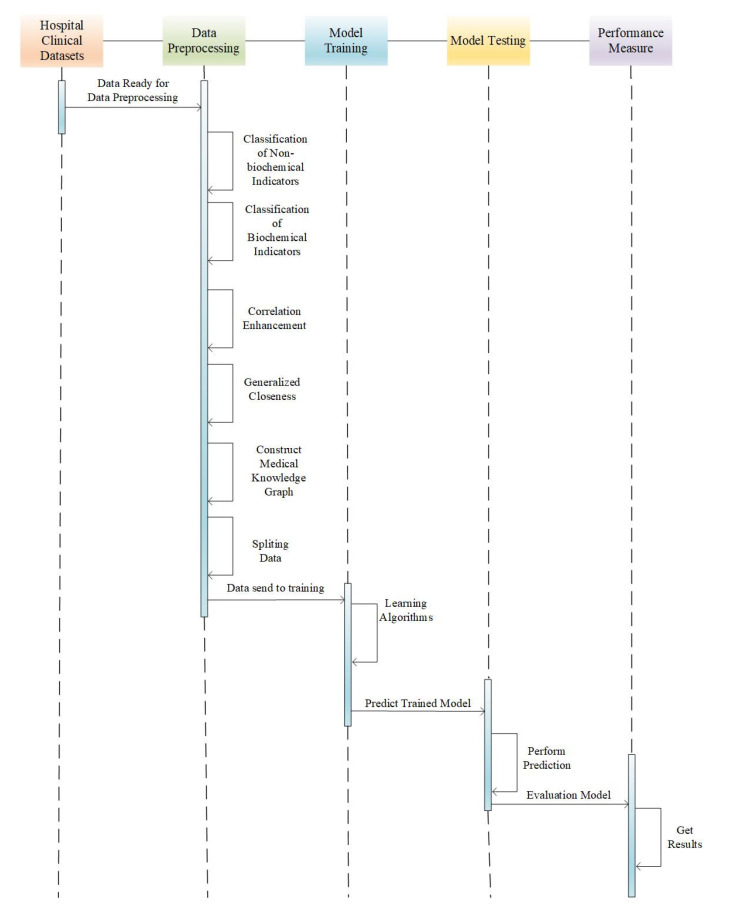
Activity sequence diagram of DME disease prediction model.

**Figure 4 diagnostics-13-01858-f004:**
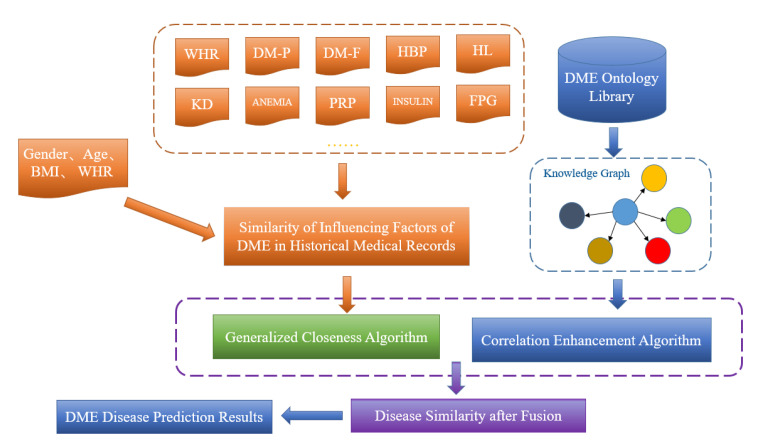
Schematic diagram of DME disease prediction model based on improved correlation enhancement.

**Figure 5 diagnostics-13-01858-f005:**
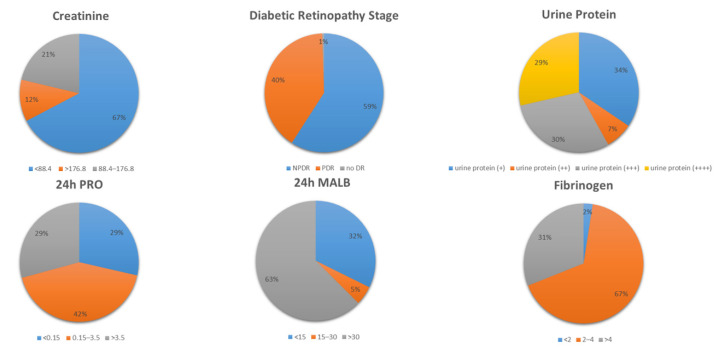
Results of statistical analysis of the six disease influencing factors of DME with the highest weight.

**Figure 6 diagnostics-13-01858-f006:**
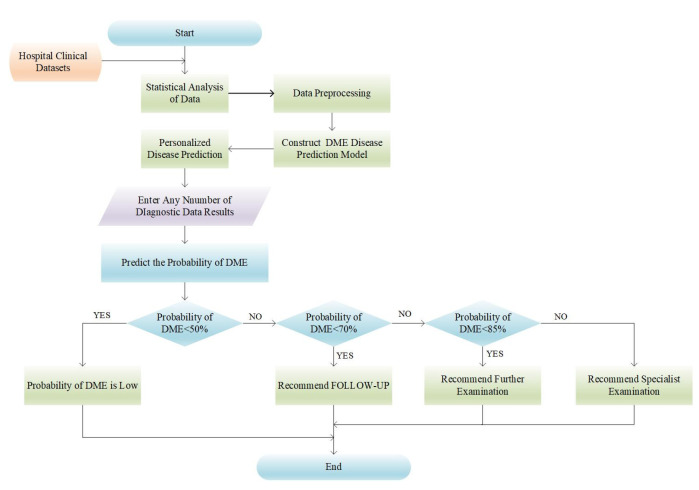
Flowchart of DME disease prediction system.

**Table 1 diagnostics-13-01858-t001:** Disease influencing factors.

Influencing Factor	Abbreviation	Unit of Measurement
Gender	--	--
Age	--	years of age
BMI	--	$kg/m^{2}$
Waist-to-hip ratio	WHR	--
Duration of diabetes	DM-P	years
Family history of diabetes	DM-F	--
Hypertension	HBP	--
Hyperlipidemia	HL	--
Diabetic nephropathy	KD	--
Anemia	ANEMIA	--
Coronary heart disease	H	--
Cerebral infarction	B	--
Diabetic peripheral neuropathy	N	--
Dark spots on the anterior tibia	DARK	--
Stage of diabetic retinopathy	DR	--
History of cataract surgery	CAT	--
History of pan retinal photocoagulation	PRP	--
History of smoking	S	--
History of alcohol consumption	A	--
Use of insulin	INSULIN	--
Fasting plasma glucose	FPG	mmol/L
Glycosylated hemoglobin	HAlc	%
Mean platelet volume	MPV	fL
Erythrocyte sedimentation rate	ESR	mm/h
Triglyceride	TG	mmol/L
Total cholesterol	TC	mmol/L
High density lipoprotein cholesterol	HDL-C	mmol/L
Low density lipoprotein cholesterol	LDL-C	mmol/L
D-dimer	DD	mg/L
Fibrinogen	FIB	g/L
Urea	UREA	mmol/L
Serum creatinine	Scr	μmoI/L
Uric acid	UA	μmol/L
Estimated glomerular filtration rate	eGFR	mL/min
Total protein	T-pro	g/L
Albumin	ALB	g/L
Urinary protein	U-pro	--
24 h urinary microalbumin	24 h MALB	mg
24 h urinary protein	24 h PRO	g

**Table 2 diagnostics-13-01858-t002:** Data processing of fasting plasma glucose data.

Plasma Glucose Range (mmol/L)	Explanation
>7	Hyperglycemia
6.1–7	Impaired fasting glucose
3.9–6.1	Normal
2.8–3.9	Normal
<2.8	Hypoglycemia

**Table 3 diagnostics-13-01858-t003:** Data processing of family history of diabetes.

Numerical Value	Explanation
0	No family history of diabetes
1	Family history of diabetes

**Table 4 diagnostics-13-01858-t004:** Classification statistics of 39 kinds of influencing factors.

Influencing Factor	Classification	Frequency	Classification	Frequency	Classification	Frequency	Classification	Frequency
Gender	male	283	female	226				
Age	≥70	63	60–69	177	50–59	154	20–49	113
BMI	<18.5	4	18.5–24	144	24–28	212	>28	147
WHR	male > 0.9	165	male 0.85–0.9	45	male < 0.85	3		
	female > 0.8	166	female 0.67–0.8	3				
DM-P	≥20 years	122	10–19 years	227	<10 years	158		
DM-F	Yes	301	No	206				
HBP	Yes	331	No	176				
HL	Yes	378	No	129				
KD	Yes	236	No	271				
ANEMIA	Yes	101	No	406				
H	Yes	327	No	180				
B	Yes	160	No	347				
N	Yes	381	No	126				
DARK	Yes	89	No	418				
DR	PDR	23	NPDR	330	No DR	63		
CAT	Yes	73	No	434				
PRP	Yes	82	No	425				
S	Yes	184	No	323				
A	Yes	135	No	372				
INSULIN	Yes	324	No	183				
FPG	>7	317	6.1–7	54	3.9–6.1	57	<3.9	5
HAlc	>9	187	6.5–9	256	6–6.5	42	4–6	21
MPV	>11	112	7–11	395				
ESR	male > 15	166	male 0–15	97	female > 20	151	female 0–20	48
TG	>2.25	155	1.7–2.25	80	0.45–1.7	270	<0.45	1
TC	>6.18	133	5.18–6.18	106	2.9–5.18	251	<2.9	16
HDL-C	>1.55	58	1.04–1.55	224	0.9–1.04	105	<0.9	119
LDL-C	>4.12	147	3.37–4.12	126	2.07–3.37	185	<2.07	47
DD	>0.5	116	0–0.5	390				
FIB	>4	90	2–4	398	<2	17		
UREA	>7.5	115	2.5–7.5	392	<2.5	0		
Scr	>176.8	26	88.4–176.8	80	<88.4	401		
UA	male > 416	67	male 150–416	212	male < 150	4		
	female > 357	68	female 89–357	156	female < 89	0		
eGFR	>120	24	80–120	353	<80	129		
T-pro	>80	13	60–80	422	<60	72		
ALB	>55	1	40–55	300	<40	206		
U-pro	urine protein (++++)	77	urine protein (+++)	101	urine protein (++)	30	urine protein (+)	299
24 h MALB	>30	196	15–30	50	<15	240		
24 h PRO	>3.5	59	0.15–3.5	110	<0.15	338		

**Table 5 diagnostics-13-01858-t005:** Correlation enhancement algorithm.

Priority	Range of Results for Weight 2	Correlation Enhancement Coefficient
1	40–60	1
2	30–70	2
3	20–80	3
4	10–90	4
5	0–100	5

**Table 6 diagnostics-13-01858-t006:** Probability of DME and recommendations.

Probability of DME (%)	Recommendations
<50	Probability of DME is low
50–70	Recommend follow-up
70–85	Recommend further examination
>85	Recommend specialist examination

**Table 7 diagnostics-13-01858-t007:** Predictions on the test set.

Probability of DME (%)	<50	50–70	70–85	>85	Total
DME	0	8	44	6	58
Non-DME	3	16	24	0	43
Total	3	24	68	6	101

**Table 8 diagnostics-13-01858-t008:** DME disease prediction results.

ID	Probability of DME (%)	ID	Probability of DME (%)	ID	Probability of DME (%)
1	74.85	21	76.84	41	73.07
2	73.37	22	61.82	42	75.50
3	84.32	23	67.73	43	90.28
4	75.67	24	71.48	44	85.01
5	73.53	25	74.91	45	75.86
6	82.36	26	76.36	46	70.13
7	77.07	27	78.53	47	75.81
8	94.25	28	75.13	48	81.53
9	72.71	29	70.16	49	76.40
10	58.43	30	71.87	50	70.50
11	76.18	31	80.50	51	73.14
12	72.50	32	74.23	52	54.37
13	68.66	33	78.75	53	90.35
14	72.01	34	81.89	54	71.08
15	85.64	35	82.23	55	57.80
16	69.20	36	89.81	56	72.18
17	73.29	37	73.81	57	71.92
18	58.92	38	70.06	58	73.38
19	80.98	39	72.69		
20	81.78	40	72.32		

**Table 9 diagnostics-13-01858-t009:** Results of evaluation metrics for link prediction.

DME Disease Prediction Model	AUC	Precision	Ranking Score
Based on statistical rule reasoning	19.61%	19.61%	0.511
Based on correlation enhancement	58.26%	57.76%	0.502
Based on improved correlation enhancement	86.21%	86.21%	0.508

## Data Availability

The data presented in this study are available on request from the corresponding authors. As the research data are confidential, the data cannot be made public.

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
