# Peer review of "Prediction of Diabetic Macular Edema Using Knowledge Graph"

_diagnostics, 2023, doi:10.3390/diagnostics13111858_

Round 1

Reviewer 1 Report

The authors proposed an improved correlation enhancement algorithm based on knowledge graph reasoning to evaluate the factors influencing DME to achieve disease prediction comprehensively. There is a certain novelty, but I have the following questions: 

1. The works of literature listed in Section 2 are mostly related to Chinese medicine. Is there any related to other countries' medicine?

2. The authors mentioned "3 DME disease prediction models" in Section 3, but there is only one model introduced in the following section. Why? Only the weights changed?

3. The test set seems imbalanced, most of the data are in 70-85 probability. Does it fit the normal distribution? I think the imbalanced test set may cause the results to be unfaired.

4. The figures in the paper are blurred, not friendly to read. There are many writing errors in the paper.

Author Response

Dear reviewer, thank you for taking time out of your busy schedule to review my paper. In view of your revision opinions, I have revised this article.

  1. Knowledge graph is widely used in the field of traditional Chinese medicine. It is widely used in clinical medicine, basic medicine and health care of traditional Chinese medicine, including semantic retrieval, intelligent question and answer, decision support, etc., which lays a solid foundation for intelligent application of traditional Chinese medicine. Most of the current disease prediction methods use data mining models for analysis and calculation, which have certain limitations. The application of knowledge graph inference is widely used in the clinical practice of traditional Chinese medicine, and can also be extended to the field of disease prediction. According to your suggestion, it has been supplemented and improved in the summary part of Section 2 of the revised manuscript.
  2. The 3 DME disease prediction models mentioned in Section 3 of the article include the DME disease prediction model based on statistical rules, based on correlation enhancement, and based on improved correlation enhancement. They are progressive and interrelated reasoning models. The disease prediction model based on statistical rule reasoning predicts the disease under the setting of weight 1. Based on the statistical rule reasoning, the weight of the influencing factors was added to the correlation enhancement coefficient to obtain the calculation result of weight 3, so as to further distinguish the correlation between the influencing factors and the disease. On this basis, a disease prediction model based on improved correlation enhancement was used to calculate the disease probability by the generalized closeness method, and combined with the recommendations of clinical experts, clinical diagnosis and treatment recommendations were given according to different disease probability intervals. According to your suggestion, the 3 disease prediction models are supplemented in Section 3.4 of the revised manuscript.
  3. The disease prediction model proposed in this paper divides the training set and the test set with a probability of 8:2, and divides the DME and non-DME data in the clinical data randomly and evenly, including 406 training sets (203 DME cases and 203 non-DME cases), and 101 test sets including 58 DME clinical data. The DME disease prediction model based on improved correlation enhancement was used to predict the disease probability of DME data in the test set, so most of the disease probability results in the test set were 70%-85%, which was consistent with the expected results. According to the results of the link prediction evaluation index, the proposed model can more accurately and effectively predict the high-risk factors and high-risk populations, and help clinical early intervention of disease risk factors. According to your suggestion, the partitioning of the dataset is supplemented in Sections 3.4 and 4.
  4. According to your suggestion, the data description and language description have been revised and improved in the revised manuscript.

Reviewer 2 Report

The report is attached.

Author Response

Dear reviewer, thank you for taking time out of your busy schedule to review my paper. In view of your revision opinions, I have revised this article.

Minor Revised:

  1. According to your suggestion, the reference order has been revised in the revised manuscript.
  2. The disease prediction model proposed in this paper was based on the existing statistical rule reasoning, and further uses correlation enhancement algorithm to extract disease risk factors with high weight of influencing factors. It is a relatively simple and general method, which can be applied to other diseases prediction, diagnosis and treatment plan recommendation of major clinical diseases and other fields. It is a method that has not been implemented in the literature at present. The accuracy of the algorithm is verified and analyzed by using the link prediction evaluation indexes such as AUC, precision and ranking score, which further demonstrates the effectiveness of the algorithm. According to your suggestion, the research significance of this paper has been supplemented and improved in Section 5 of the revised manuscript.

Major Revised:

  1. According to your suggestion, the relationship between Equation 5 and the data in the paper has been supplemented in the revised manuscript.
  2. According to your suggestion, Table 7 has been moved to the appendix in the revised manuscript, and the discussion part has been explained in the article.
  3. This paper proposed a disease prediction model based on improved correlation enhancement based on diabetic macular edema disease. The knowledge reasoning method used is based on statistical rule reasoning, the weight of the connection relationship between the disease node and the influencing factor node in the knowledge graph is weighted by the correlation coefficient, and the generalized closeness formula is used to calculate the disease probability. It can not only realize the personalized disease probability prediction, but also analyze the high risk factors of disease, providing reference value for clinical application. Because the method used is the application analysis of knowledge reasoning based on knowledge graph in clinical disease prediction, the use of traditional probability analysis model fails to achieve the effect of clinical personalized disease prediction, and can not meet the disease prediction of some missing data in clinical data. The method implemented in this study can solve this problem. In the future, more data mining models will be used for model optimization and verification after more clinical data are collected. According to your suggestion, the application scenario of the algorithm and the further optimization of the probability analysis model are supplemented in the discussion section of the revised manuscript in Section 5.

Reviewer 3 Report

This manuscript tried to predict DME using knowledge graph. Overall, the conception of the study design is well documented. In fact, I don't have much to add to the study design or the construction of the paper. But there are still some minor grammar error needed to edit. Please have some native English speaker revise it to make it easier to read.

Author Response

Dear reviewer, thank you for taking time out of your busy schedule to review my paper. In view of your revision opinions, I have revised this article.

According to your suggestion, the grammar content of this paper has been revised and perfected more professionally in the revised manuscript, so as to facilitate the reading of the article.

Round 2

Reviewer 1 Report

The authors replied all my questions.

Reviewer 2 Report

The authors have revised the article nicely and can be accepted in present form.